

# Wave model verification based on measurements in the Wadden Sea

Cordula Berkenbrink[1], Luise Hentze[2], Andreas Wurpts[1]

[1]Coastal Research Station (NLWKN), Norderney, 26548, Germany
[2]Franzius Institute (Leibniz University Hanover), Hanover, 30167, Germany

*Correspondence to*: Cordula Berkenbrink (cordula.berkenbrink@nlwkn.ny.niedersachsen.de)

**Abstract.** The design height of coastal protection structures in Lower Saxony / Germany is determined by the design water level and the corresponding wave run up. For the calculation of these parameters several mathematical models are used which need to be verified for the conditions at the East Frisian Wadden Sea area. For this issue a wave measuring programme is operationally run, which includes various measurement locations and devices around the islands Norderney

and Juist. The measurements are continuously extended and adapted in order to improve models and measurements.

This paper shows a comparison between measured and calculated data for the storm surge of the 10.-11.01.2015 incorporating to new wave and water level gauges operated within COSYNA as well as a second research project dealing with wave attenuation behind barrier islands. Water levels within the investigation area were calculated by hydrodynamic models driven with a wind field originating from weather forecast and compared to water level measurements. The

corresponding wave energy field was calculated by means of a third generation wave model and results compared to measurements of several devices located around the barrier Islands. The aim of the study shown here is to give a brief overview of possible error sources for model-data as well as data-data comparisons.

## 1 Introduction

The Coastal Research Station is involved in the design of coastal protection structures in Lower Saxony – Germany. In

respect of both safety and economic efficiency the main tasks are the evaluation of water levels, wave parameters and the verification of mathematical models to ensure, that the design is done by state of the art considering the valid law. Since the seventies, a measuring programme around Norderney Island is continuously operated, which is adapted to the various issues. Some locations are fixed for generating long time series of wave measurements, other locations are flexible.

The measuring programme is used for validation and calibration of mathematical modelling of the wave climate, especially

for extreme conditions like storm surges. New numerical or physical approaches can be proved easily by means of model and measurement comparison.

The wave climate directly depends on the water level, the wind and the topography, which means that these boundary conditions must be known as detailed as possible. The difficulty is to reproduce the water level in the investigation area, which has an increasing gradient from offshore to onshore in case of storm surges. In 2012 the measuring programme was

extended with two WaveGuide radar distance sensors made by Radac Company. They are part of the monitoring project



COSYNA (Coastal Observing System for Northern and Arctic Seas) and installed, operated and maintained by the Coastal Research Station. One position is located in the Wadden Sea close to Norderney tidal inlet (location Steinplate), the other one is located close to Ley Bay (location Bantsbalje).

## 2 Investigation area

The modelling was done in the area around the island Norderney in the Wadden Sea in the northern part of Germany where validation data from several measurement systems are available (Figure 1). Directional wave rider buoys made by Datawell Company records the wave climate in the North Sea. The tidal inlet and the Wadden Sea are equipped with non-directional wave rider buoys also made by Datawell. At the western boundary a buoy produced by Axys Company measures the wave climate of the Osterems. Additionally, several poles equipped with radar or ultrasonic wave sensors record water level and
wave climate.

## 3 Method

Tide and wave computations were carried out on an existing curvilinear grid model basically covering Norderney tidal inlet an including the measuring area. The open boundaries are located at buoy positions (locations OEMS-N and FW). The bathymetry is based on laser scan data and echo soundings from the years 2010 to 2013. The water level was modelled with
the hydrodynamic model Delft3D under astronomical tide and wind conditions of the evaluation period. Thereafter the wave climate is modelled with the third-generation wave model SWAN (Ris et al. 1995, Booij et al. 1999, Holthuijsen et al. 1998). As a first step this was done in a semi-coupled and stationary mode and compared to measured data.

## 4 Boundary Conditions

### 4.1 Wind

The wind field numerical dataset was provided by the German National Meteorological Service. The used prediction model COSMO-EU (Consortium for Small Scale Modelling - Europe) calculates, inter alia, winds speed and direction for the European continent with a resolution of 7 kilometers. Compared to the measurements the predicted data for direction is in the same range, but the wind speed is more constant in the prediction model for the island of Norderney (Figure 2). The period of high wind speeds is longer in the prediction model, but the maximum values have not been reached.

### 4.2 Water level

The water level in the investigation area depends on the astronomical tide and the wind shear at the water surface induced by especially stronger wind events over the North Sea. For getting realistic water level time series, the storm surge was



reproduced for the whole continental shelf by means of the hydrodynamic Delft3D model forced by a Smith and Banks type wind shear boundary condition and pressure field. In a cascade of nested models (Figure 3) with increasing resolution the water level was calculated for the investigation area by using the COSMO-EU wind field and the astronomical tide for a time period of one week.

The results are compared with recorded data sets from the radar distance sensors of the COSYNA - Project (Figure 4). The calculated water level differs from the measured one basically in the same way as the wind forecast differs from the recorded values. Where the predicted wind field agrees to the measured data set, the calculated water level fits to the measured one. If the prediction exceeds the measured signal (Figure 2), the calculated water level exceeds the recorded one (Figure 4) and the

other way around. The deviation is induced by the predicted wind field which wasn't adjusted by the authors, since the maximum water level used for the stationary calculation of the wave climate is sufficiently accurate. Also, during the maximum water level period the stationary and one-way coupled wave model approach is considered sufficiently realistic, since current velocities during the storm surge peak drop to very small values. The chosen period for the stationary wave modelling is the 11.01.2015 02:00. The water level computed by Delft3D is passed to the wave model SWAN (Figure 5).

**4.3 Wave climate at the boundaries**

The northern offshore boundary of the wave model runs in east-west direction, roughly along the edge of the fairway. At location FW there is a wave rider buoy measuring the wave climate. This dataset is used as the input signal for the whole northern boundary (Figure 6). For the western boundary a buoy of the Osterems (location OEMS-N) serves at the basis for the boundary. The eastern boundary is free just like the southern boundary towards the Ems. The rest of the southern

boundary is closed by the dike line.

**5 Wave modelling and comparison with measurement data**

An exemplary comparison of measured and calculated wave spectra and parameters is presented in order to highlight the achievable accuracy of wave modelling under storm surge conditions in the coastal zone and the Wadden Sea area. From the position FW the energy density decreases towards the shore. Although the buoy at position FW serves as boundary condition

calculation and measurement do not coincide exactly (Figure 7). The significant wave height is overestimated by 5 % and the energy period by 2 %. Even though the energy density at the next position (SEE) is underestimated by the mathematical model, the significant wave height is about 9 % too low. The energy period is also underestimated by 5 %. The shape of the modelled spectrum is similar to the measured one which means, that the physical processes are well reproduced. An edge borders the reef seaward the island where the buoy is located, an exact reproduction with the curvilinear grid can't be

guaranteed. The magnitude and the shape of the wave spectra change behind the reef. The dominant peak of the long waves reduces by 95 % in the measurement values towards position VST. For this position there are long measurement time series



available. In 2013, where the model bathymetry dates from, the measured wave spectra show peaks in comparable scale like the model calculated at this position. This is also confirmed by previous studies (Kaiser and Niemeyer 2001). We assume that the usually very dynamic bathymetrical changes at the ebb tidal delta and foreshore are the cause for this behaviour. The correlation between ebb tidal delta morphology and wave climate for continuously monitoring of the riff by means of

Doppler radar measurements is currently under investigation (Flampouris et al. 2008).

The wave propagation behind the barrier Islands will be shown exemplary for the area behind the island of Juist, where data of several measuring systems where available (Berkenbrink et al, 2016). The different measurment devices show plausible results. The wave energy propagates through the tidal inlet while it converts energy towards higher frequencies, the further it goes into the Wadden Sea (Figure 8). The same general behavior is found in the mathematical model results, but on an lower

overall energy level. The significant wave height is about 10 % lower while the energy period is about 10 -15 % lower than observed.

At the position NL it can be seen that grid resulution behind the island of Juist is not sufficcient: There are two measurement positions in close vicinity to each other which are used to assess the strong local wave attenuation gradient within a separate research project (Berkenbrink et al., 2016). The grid used here doesn't provide enough grid cells between locations NL-S

and NL-N in oder to achieve the observed dissipation in the modelled wave spectra.

The deviation between measured and calculated wave spectra at Bantsbalje is quite high, and it can be seen, that the radar distance sensor at position Bantsbalje records more wave energy compared to the bouys and the hydroaccoustical sensors upstream at locations KS and NL. This physically unrealistic relation is currently investigated by direct comparison between the radar distance sensor and a wave bouy.

**6 Conclusion and Outlook**

The paper shows a model to measurement comparison for a complex topographical setting at two tidal inlets of the East Frisian Wadden Sea coast. Deviations between model and measured data for a storm surge case are briefly evaluated and qualitatively interpreted with respect to different error sources. The need for several further investigations is shown.

The quality of measurement results not only differs between buoys and poles but also between the buoy types themselves,

why further investigations are needed for verification and comparison of the different measurement systems in the Wadden Sea. A verification test with three buoy systems seawards of Norderney is scheduled for summer and autumn 2016 in cooperation with Axys Company, another one will be carried out at the inner tidal ebb delta of Osterems (OEMS-N). The wave spectra of position "Bantsbalje" needs further investigation. The same applies to the position "VST", where new bathymetry data are necessary.

The model investigation also shows that apart from easy to identify dependencies between calculated wave parameters and bottom topography also sufficient resolution is required in areas of strong gradients in the numerical solution. The latter cannot as easily be identified a priori and might requires consecutive model runs with locally improved resolution in order to



get realistic results. This clearly points towards the use of unstructured model grids in tidal inlets. Going beyond the simplified approach shown here, non-stationary calculations will provide deeper insight into the temporal development of the observed parameters. An important prerequisite though is a good agreement between the driving wind field and the corresponding wind measurements.

**7 Acknowledgements**

The work presented here is part of the Bachelor Thesis of Luise Hentze which was submitted at Franzius Institute of Hanover University and conducted at Coastal Research Station of the Lower Saxony Water Management, Coastal Defence and Nature Conservation Agency. The presented data of the measuring positions "Steinplate" and "Bantsbalje" are part of the COSYNA-Programme. They are online available at: http://codm.hzg.de/codm/. The measurement devices behind the
island of Juist are partly funded by the German Federal Ministry of Education and Research (BMBF) within the framework of the programme of the German Committee on Coastal Engineering Research (KFKI) - project code 03KIS101.

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





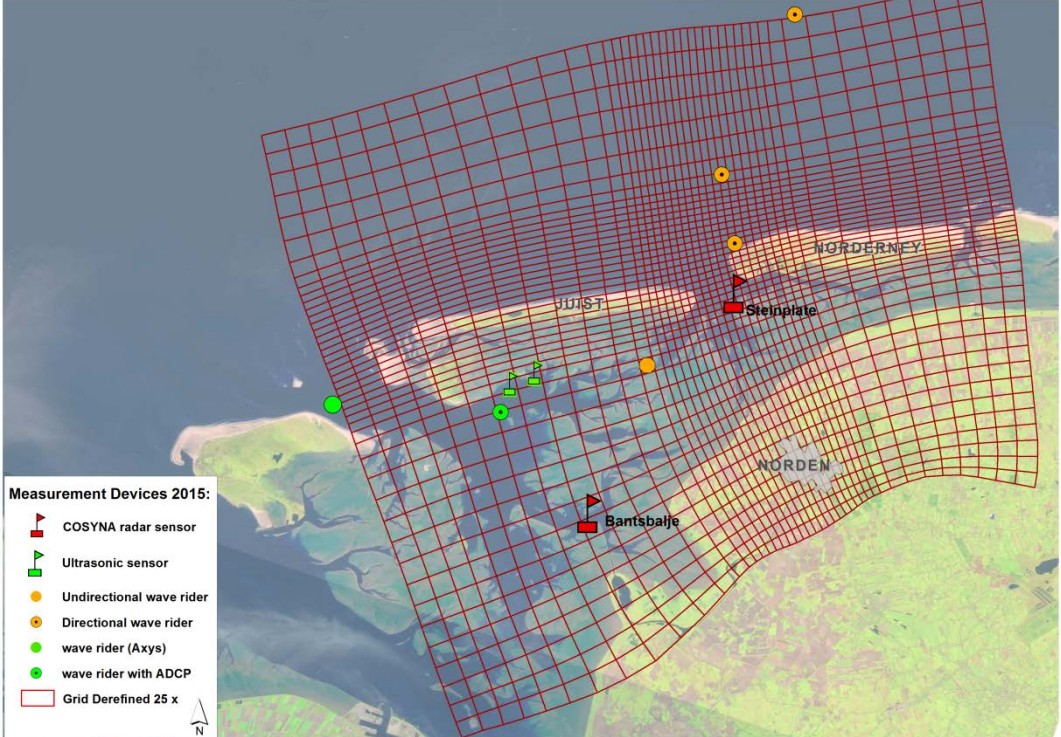

**Figure 1: Investigation area: location of measurements devices and orientation of the model grid.**



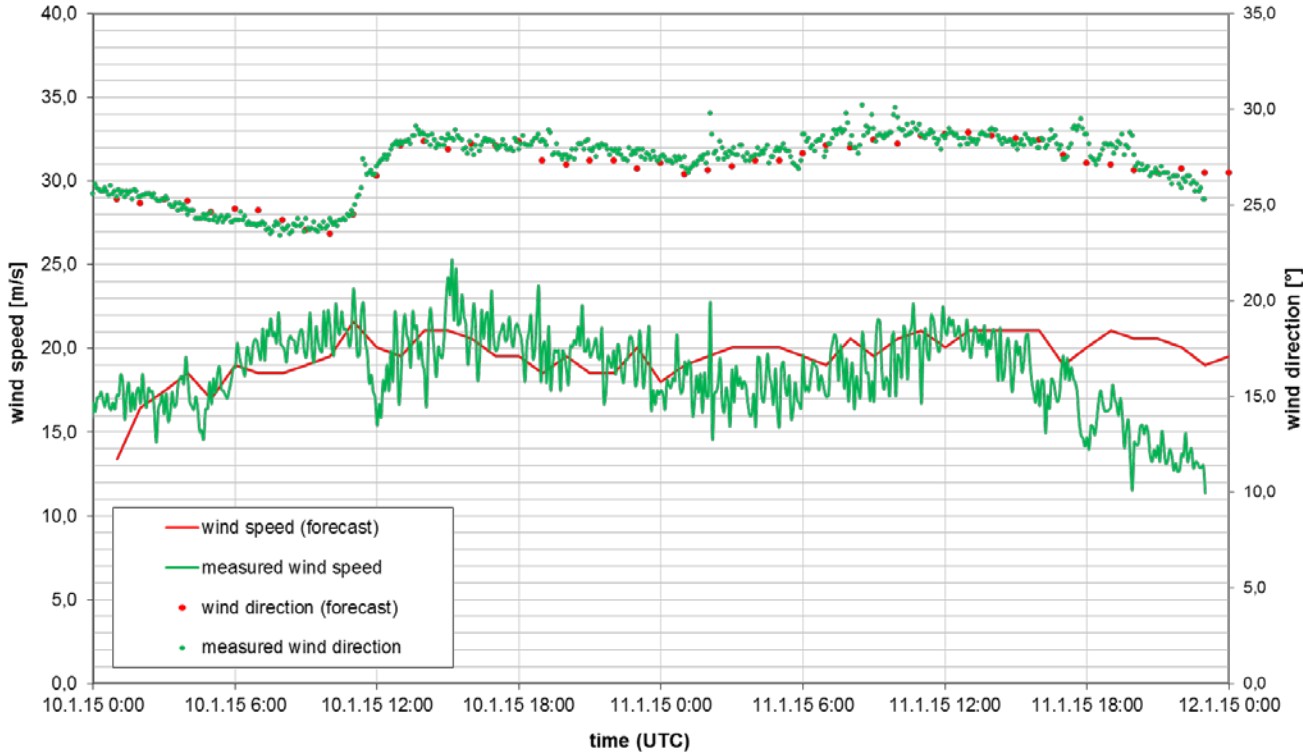

**Figure 2: Comparison of predicted and measured wind speed and direction for the island of Norderney. Data interval of measured values: 5 min, calculated values: 60 min.**





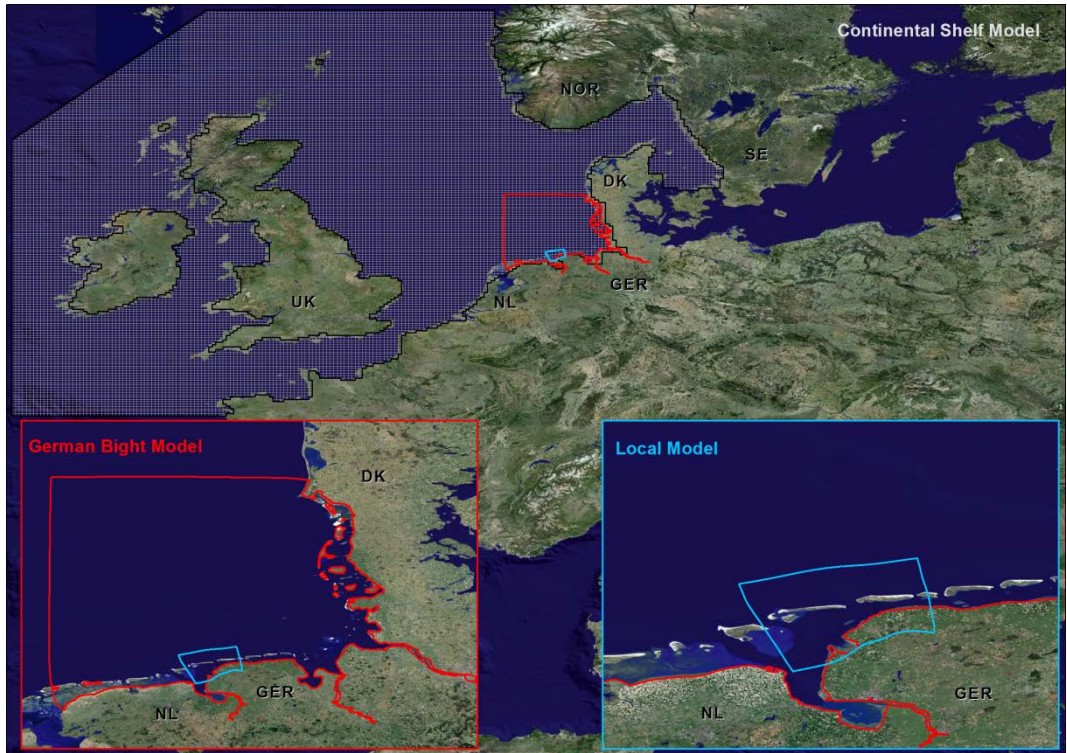

Figure 3: Cascade for storm surge modelling.





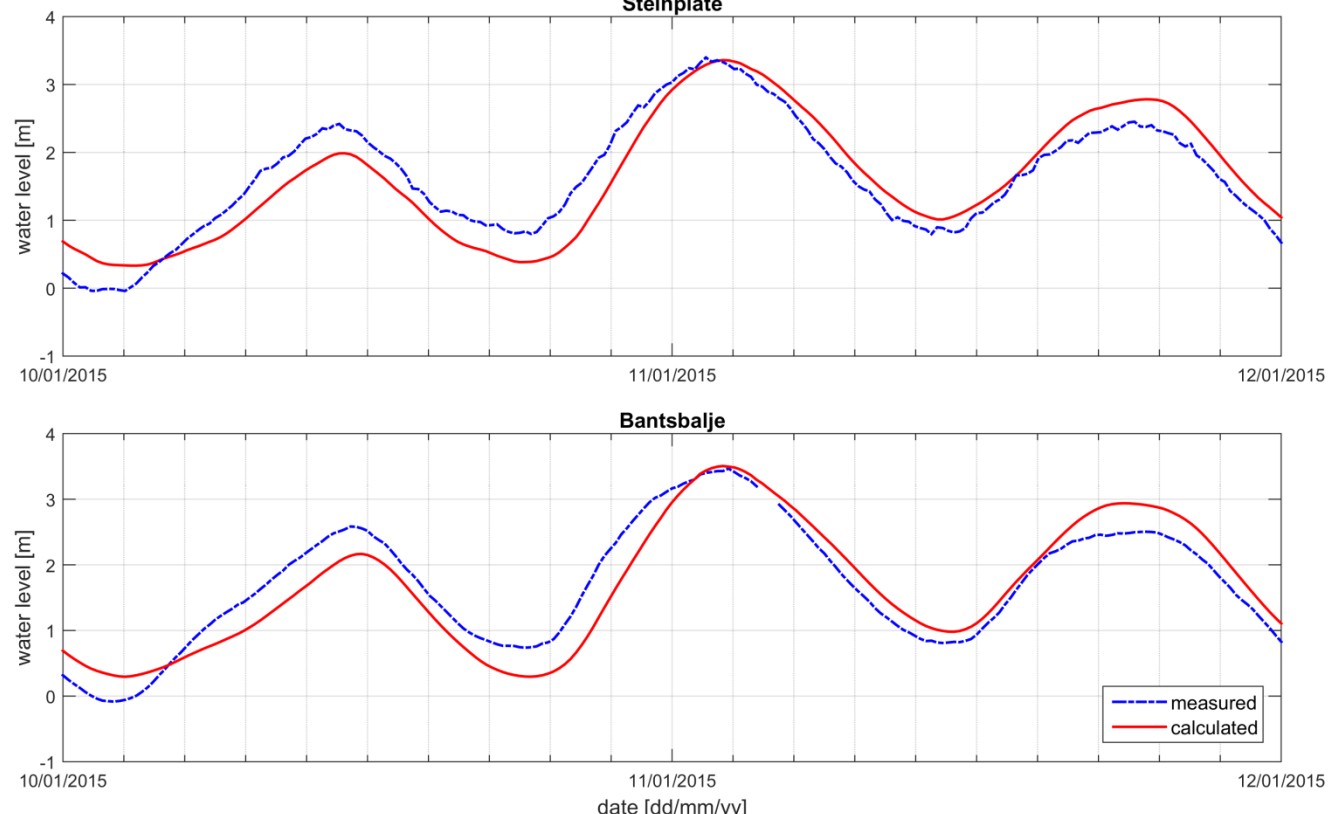

**Figure 4: Comparison of measured and calculated water levels at the radar positions.**




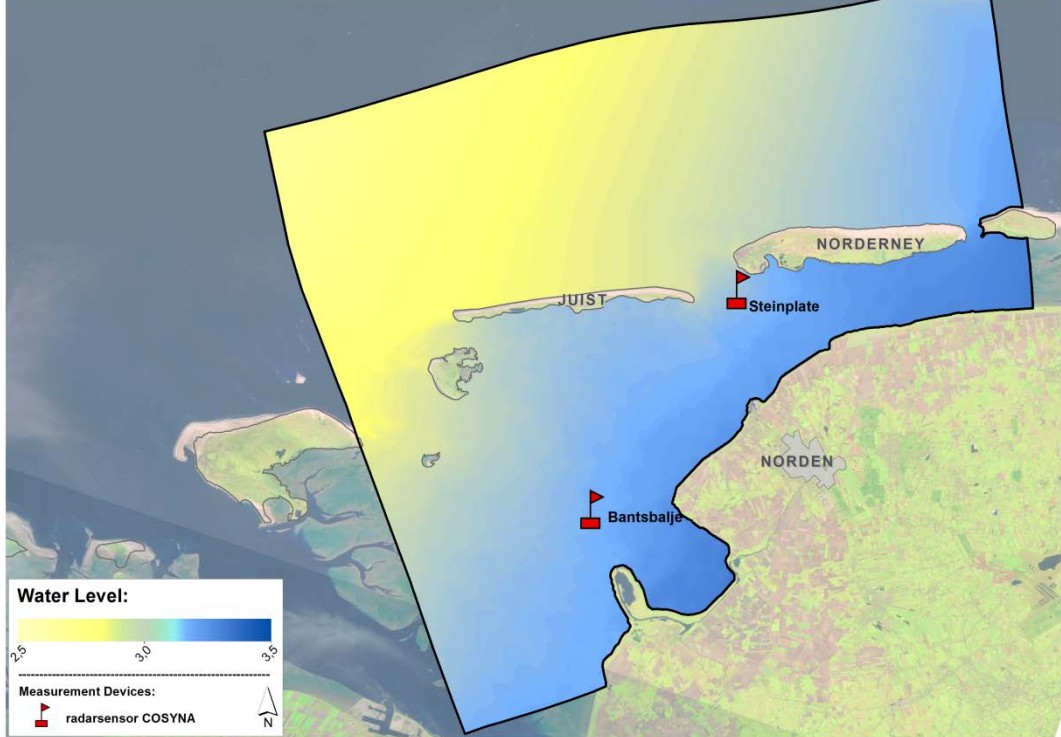

**Figure 5: Water level in the investigation area at 11.01.2015 02:00.**





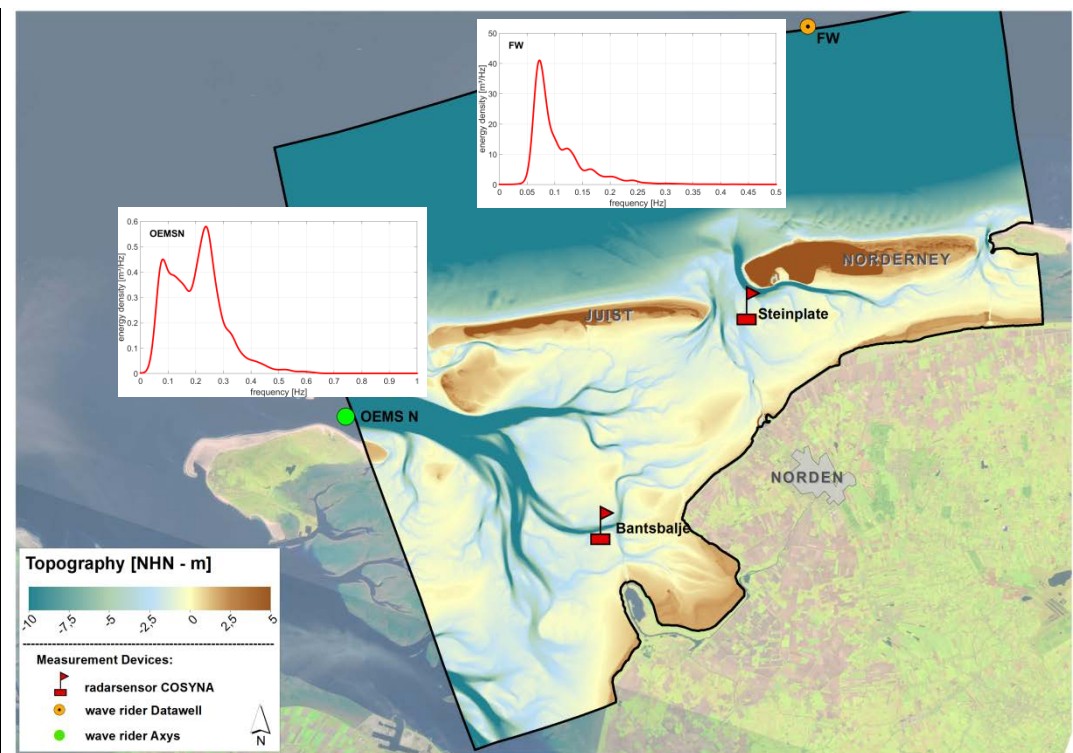

Figure 6: Wave spectra at the boundaries.





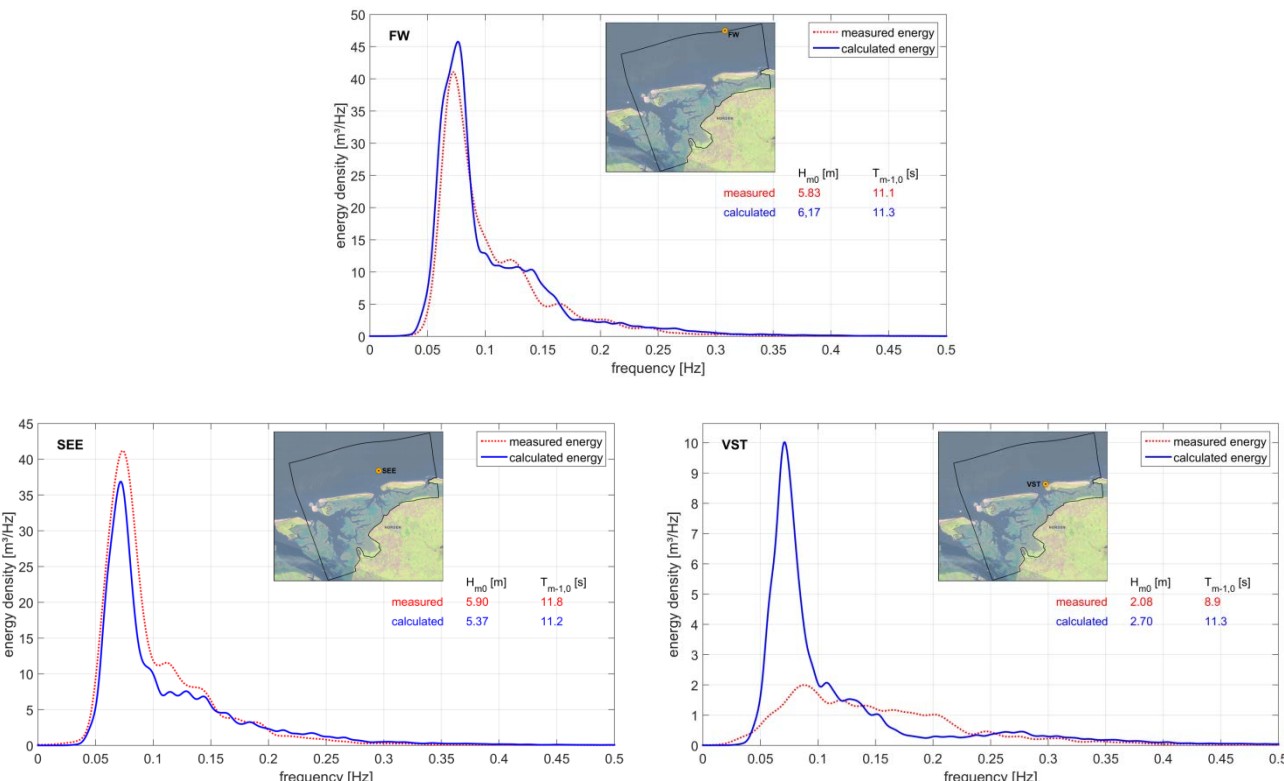

**Figure 7: Model verification for the coastal zone area.**




Figure 8: Model verification for the tidal inlet and Wadden Sea.