# Peer review of "Wave model verification based on measurements in the Wadden Sea"

_Ocean Science, 2016_

## Referee Comment (RC1) · Anonymous Referee #1 · 3 Jun 2016

"Wave model verification based on measurements in the Wadden Sea"

The authors describe a SWAN-Delft3d model for a region offshore of Germany. Water levels offshore appear to be relatively accurate. Wave predictions on the other hand appear to have significant errors in the nearshore.

Overall Impression

In general the paper lacks sufficient detail. There are many different physical parameters that are typically adjusted in models like SWAN and Delft3d. The authors discuss none of these. Additionally boundary conditions, in particular the SWAN model, needs far more detail. Wave observations from directional buoys are non-trivial. The assumption of homogeneity along the boundary should also be addressed. Additionally I found that figures lack appropriate details and are generally redundant.

The aim of the paper is to discuss the errors, however it lacks discussion of physical and numerical possibilities. There is a rich literature on wave dissipation due to bottom friction and white-capping, this should be discussed and would make the paper much stronger. See for example many papers by Fabrice Ardhuin, and the state of the art paper on wave modeling by Cavaleri et al (2007). As it is, the paper does not add any insights to the current literature. I believe it could add to our understanding of wave processes in the nearshore if the authors include significantly more discussion and investigation of errors in context of numerical and physical deficits, as well as model setup and forcing. Dissipation settings could be adjusted in SWAN, model spatial resolutions could changed and compared, and boundary forcing could be more carefully prescribed.

General Comments

The language is wordy, sometimes unclear, and would benefit from some revision by a native English speaker. Words such as "just" and contractions "wasn't" should be removed.

The authors include plan view figures of the domain in many figures (1,3,6,7,8). This is unnecessary. Additionally all plots need x/y labels, e.g. Lat/Lon. Many readers may be unfamiliar with the domain, some description in the text with appropriate figure would be useful.

It may be owing to differences in scientific communities, however I prefer using predicted/modeled instead of calculated for referring to model output. Calculated seems ambiguous.

Section Specific Comments

Section 2 Include more detailed about buoy model numbers, e.g. MKIII, as well as mooring times and lat/lon locations. All instrument locations should be clearly labeled on a plot like Fig 1.

Section 3 Do SWAN and Delft3D use the same grid? What specific model settings were used? E.g. what dissipation settings are used in SWAN? Is diffraction used? In stationary mode wind-wave generation can be overestimated if actual winds are non-stationary, and do allow for fully developed seas. The authors should discuss this and other model settings in detail. Ideally enough details should be included to replicate the study. Additionally the model coupling should be discussed futher, one-way, two? Examining spectra in Figure 7 makes we wonder whether the authors included triad interactions in their SWAN model.

Section 4 4.2 – Is there a citation for this Delft3D simulation? If not, can more detail be included as to how this nested model was run and forced?

"The calculated water level differs from the measured one basically in the same way as the wind forecast differs from the recorded values." - This is not obvious. Combine plots to illustrate? And/or examine correlations. 4.3 – Directional wave buoys do NOT simply measure E(f,theta) needed to force a wave model such as SWAN. The following is not sufficient "At location FW there is a wave rider buoy measuring the wave climate. This dataset is used as the input signal for the whole northern boundary" The authors need to specify what wave parameters were used to force the boundary, bulk parameters, or E(f,theta). See Longuet-Higgins et al 1963, Ochoa & Delgadogonzalez 1990, Long 1980

Additionally station OEMS-N appears to be sheltered by a large number of directions. Is this accurate for the entire western boundary?

Section 5 Line 26 – Does mathematical model refer to SWAN? Line 26 – The under-estimation of the spectra at SEE could mean the model is missing dissipation. Please comment on whether this may be due to white-capping or bottom roughness.

Line 30 onward – It is not clear the physical reasons for the severe model inaccuracy at site VST. Please re-write.

Specific Comments Figure 1: Needs x/y axis. Needs colorbar

Figure 3: Again, x/y axis needed. Additionally Insets are hard to separate from image, consider different bordering, or using contours instead of color. Caption should be more specific, Nested grids A, B, C, etc. with resolutions XX, or spatial extents XXX

Figure 5: Again x/y axis needed. Region outside of domain is similar color to colors in colorscale, set to white or another color to make it clear this is not modeled. Caption should say modeled water level to distinguish between observed modeled.

Figure 6: Again x/y axis needed. Caption does not describe figure. Figure shows bathymetry of region. Maybe include bathymetry clearly in Figures 1 or 3. Wave spectra inset plots are too small to see clearly. Make it clear whether these are measured or modeled. Include these labels in Figure 1.

---

## Referee Comment (RC2) · Anonymous Referee #2 · 22 Jun 2016

The paper compares hydrodynamic and wind wave data collected in the East Frisian Wadden Sea coast with numerical results obtained using Delft3D and SWAN. The manuscript seems conceived much more like a "short" internal report rather than a scientific paper. The numerical models are only mentioned while they are not presented and discussed. Although Delft3D and SWAN are well-established models in the scientific community the authors should, at least, discuss on the parameters they use when preforming their simulations. In principle, the reader should be able to perform the same computations by himself but, in this case, the lack of information makes this unfeasible. Very few details the authors provide also for the data. The agreement between numerical results and measured data is not good in many cases and the discussion about the reasons for these discrepancies is only superficial. It is not clear to me the "scientific" message of this contribution apart from "The need for several further

investigations" as the authors clearly state in the conclusion section. Furthermore, I cannot say the paper is well written and English is often poor. Concluding I cannot recommend the manuscript for publication in OS in the present form. All the same, I provide a set of comments and suggestions.

SPECIFIC POINTS:

Page 1 line 8: "a wave measuring programme is operationally run". Which is the program? It is COSYNA that you mention later in the abstract?

Page 1 line 12: "COSYNA" is an acronym. You define it later in the text (Page 2 Line 1) but you should define it the first time you use it.

Page 1 line 19: Coastal Research Station should be identified more clearly.

Page 2 line 3: I suggest mentioning fig 1 here.

Page 3 line 7: "the wind forecast differs from the recorded values". Given that you are simulating an event from the past, why don't you use hindcasting reconstruction to force you model? Page 3 line 11: "during the maximum water level period the stationary and one-way coupled wave model approach is considered sufficiently realistic, since current velocities during the storm surge peak drop to very small values". This statement should be supported in a quantitative way.